# Functional Characterization of the N-Terminal Disordered Region of the *piggyBac* Transposase

**DOI:** 10.3390/ijms231810317

**Published:** 2022-09-07

**Authors:** Gerda Wachtl, Éva Schád, Krisztina Huszár, Antonio Palazzo, Zoltán Ivics, Ágnes Tantos, Tamás I. Orbán

**Affiliations:** 1Institute of Enzymology, Research Centre for Natural Sciences, Eötvös Loránd Research Network, 1117 Budapest, Hungary; 2Doctoral School of Biology, Institute of Biology, ELTE Eötvös Loránd University, 1117 Budapest, Hungary; 3Department of Biology, University of Bari “Aldo Moro”, 70125 Bari, Italy; 4Transposition and Genome Engineering, Division of Medical Biotechnology, Paul Ehrlich Institute, 63225 Langen, Germany

**Keywords:** DNA transposon, *piggyBac*, PGBD, intrinsically disordered protein

## Abstract

The *piggyBac* DNA transposon is an active element initially isolated from the cabbage looper moth, but members of this superfamily are also present in most eukaryotic evolutionary lineages. The functionally important regions of the transposase are well described. There is an RNase H-like fold containing the DDD motif responsible for the catalytic DNA cleavage and joining reactions and a C-terminal cysteine-rich domain important for interaction with the transposon DNA. However, the protein also contains a ~100 amino acid long N-terminal disordered region (NTDR) whose function is currently unknown. Here we show that deletion of the NTDR significantly impairs *piggyBac* transposition, although the extent of decrease is strongly cell-type specific. Moreover, replacing the NTDR with scrambled but similarly disordered sequences did not rescue transposase activity, indicating the importance of sequence conservation. Cell-based transposon excision and integration assays reveal that the excision step is more severely affected by NTDR deletion. Finally, bioinformatic analyses indicated that the NTDR is specific for the *piggyBac* superfamily and is also present in domesticated, transposase-derived proteins incapable of catalyzing transposition. Our results indicate an essential role of the NTDR in the “fine-tuning” of transposition and its significance in the functions of *piggyBac-*originated co-opted genes.

## 1. Introduction

Transposons are mobile genetic elements capable of relocating from one genomic locus to another, and they often make up significant portions of the genomes of organisms, including humans [1]. There are two classes of transposons: members belonging to the RNA transposons (or retrotransposons) use an RNA intermediate for their replication cycle and use a ‘copy-and-paste’ mechanism for transposition. In contrast, the class of DNA transposons contains elements that rely on DNA intermediates for their propagation, and the reaction mainly occurs via ‘cut-and-paste’ mechanisms [2,3]. Transposons were initially considered mutagenic, “selfish” genomic parasites. However, they could also contribute to the adaptive evolution of organisms by either carrying advantageous genes for survival (such as antibiotic resistance genes for bacteria) or by becoming harmless “domesticated” genes that serve a novel endogenous function of the host [4,5,6,7,8,9,10].

Due to their simple structure and the cut-and-paste mechanism of non-replicative nature, some DNA transposons became favorable genetic tools for mutagenesis or gene delivery experiments [11]. For mammalian applications, the insect-derived *piggyBac* (PB) [12,13], the resurrected artificial *Sleeping Beauty* (SB) [14], and the medaka fish-originated *Tol2* [15,16] transposons are all widely used, the two former ones being the most favorable systems, partly because of the development of their high efficiency (“hyperactive”) variants [17,18]. The SB system has a clear advantage for human gene therapy applications due to its safest integration profile. Among all integrating gene delivery vehicles, SB has the least tendency to integrate into endogenous genes, providing the lowest likelihood for a genotoxic event. In addition, the lack of potentially cross-reacting endogenous elements makes it an attractive choice for human applications [19]. Considering the PB system, there are PB-like elements in the human genome, and their cross-reactivity with the insect transposon is still a matter of debate [20,21,22]. On the other hand, the PB transposase performs a seamless excision of the transposon unit from a genomic location [23,24,25]. This ability to remove the DNA substrate without leaving any footprints is convenient for specific applications: a prominent example is genetic reprogramming, when the transgenic cassette can be safely eliminated after the desired cellular phenotype has been achieved [26,27]. Apart from exploiting the beneficial characteristics of a given transposon system, there are continuous efforts to further optimize the transposase enzymes for more efficient and controllable reactions [28]. To achieve that, there is also a constant need to understand the structure-function correlations of the transposases and, thereby, understand the transposition reactions in more detail.

Our studies focus on the functional characterization of the PB transposase and its relatives. They represent a widespread superfamily of DNA transposons [29,30,31] and contain active mobile elements in various genomes, including bats [32]. The first discovery of a PB transposase was made in the cabbage looper moth (*Trichoplusia ni*) cells [33,34], and this insect-derived autonomous element was proven to be functioning and applicable in various other organisms, including human cells [12]. When characterizing its structure, this transposase was shown to have a domain with an RNase H-like fold containing a DDD motif [35]: this triad of aspartates is responsible for catalyzing the DNA cleavage and joining reactions [36], similarly to retroviral integrases and the IS4 family of insertion sequences [37,38]. This catalytic domain is located in the central part of the protein. Though, as with other DDE/D structures, a small insertion domain interrupts the middle portion [39]. The catalytic domain is flanked on both sides by regions creating the ‘dimerization and DNA-binding domain’, which is responsible for bringing two proteins together to form a dimer, and also for interacting with a short sequence part of the transposon terminal inverted repeats (TIRs), as well as with the target DNA [39]. A cysteine-rich domain (CRD) is found at the C-terminal end of the protein, which contains a nuclear localization signal [40]. The CRD was proposed to be indispensable for transposon DNA binding and DNA breakage [41], and it was shown that CRDs of two interacting transposases induce the formation of an asymmetric protein dimer structure associated with a single (the left) TIR of the transposon [39]. As opposed to this, a subsequent study challenged these results by claiming that the CRD is not required for PB DNA transposition, as CRD-deficient transposases, including a domesticated PGBD5 transposase, were found to be functional [42]. A possible resolution for this issue could lie within the evolution and structure–function variability of the CRD in active and domesticated PB transposases. A comparative analysis of various cysteine-rich domains revealed significant structural differences among distantly related PB elements. As a functional consequence, it was shown that the CRD of the domesticated *piggyMac* does not bind DNA [43].

Although numerous investigations have been published on addressing the structure and function of the insect PB transposase, our knowledge about the potential role of the N-terminal region is still very limited. This ~100 amino-acid-long portion is predicted to have a highly disordered structure, and the dominantly acidic nature of the residues makes it unlikely that it contributes to DNA binding [39]. As one of the most common functions of disordered protein regions is establishing inter- or intramolecular interactions, which often provide the basis for allosteric regulation [44], it is thus conceivable that similar modulation of the PB protein is achieved by protein partners interacting with the N-terminal part of the transposase. Therefore, this study aimed to understand how this N-terminal disordered region (NTDR) of the PB transposase contributes to PB transposition. By deleting this N-terminal segment, we could detect a significant drop in overall transposition efficiency; however, the extent of the reduction showed strong cell type dependency. We also provide evidence that the excision step of transposition is heavily influenced by NTDR deletion. When the NTDR was replaced with a disordered segment that contained the same amino acids in a randomized order, wild-type transposition efficiency was not restored; in fact, the impairment was even more pronounced, indicating the importance of the protein sequence, not only the structure or the amino acid composition of this region. With a systematic bioinformatic analysis, we also show that the NTDR is specific for the *piggyBac* superfamily of DNA transposons and that it can be found even in domesticated, transposase-derived proteins that lost their ability to mobilize DNA. These include the five human *piggyBac*-derived sequences (PGBD1-5), which have been shown to have lost their mobilizing activity [22]. These results support the hypothesis that the NTDR of PB has an important regulatory function in transposition and that this regulatory function has likely been preserved during the evolution of PB-derived genes.

## 2. Results

### 2.1. Deletion of NTDR Significantly Decreases PB Transposition in a Cell Type-Specific Manner

We began our investigations using the mammalian codon-optimized version of the *T. ni* PB transposase (also called mPB, [13]) and its two-component vector system (Figure 1A). To test the influence of the NTDR on transposition, we deleted 100 amino acids from the N-terminus of the mPB transposase. This peptide region is predicted to have an intrinsically disordered segment (Figure 1B). The length of the deletion was determined by our earlier predictions showing that several PB elements have at least a 100 amino acid long disordered region at their N-termini. The resulting mutant transposase (mPBdel) was co-transfected with a PB transposon vector carrying a puromycin resistance gene cassette into HEK-293 cells. At 48 h post-transfection, transposon excision was checked by a diagnostic PCR, whereas determining the colony numbers after two weeks of puromycin selection provided information about the overall transposition efficiency of the mutant. The wild-type mPB transposase was used in parallel transfections as a positive control. In contrast, experiments with a catalytically inactive mutant (mutPB, see Figure 1B) served as a negative control to estimate the background of random integration. These assays revealed the overall proficiency of mPBdel to excise PB transposons (Figure 1C); however, the transposition rate dropped significantly, to approximately 55% of the level detected for the wild-type mPB transposase (see the colony assays, Figure 1D).

To address if the effect of NTDR deletion on mPB transposition can also be detected in other cell types, we performed similar experiments in HeLa cells. We measured a significant decrease in the overall transposition activity of the mPB-del mutant (Figure 2A,C). However, the magnitude of reduction was much higher than in HEK-293 cells: the colony numbers detected for the mPBdel reached only 20% of that measured for the wild-type PB transposase. We also tested the mPBdel activity in MCF-7 cells to further analyze transposition’s cell type specificity. The tendency was similar in that the efficiency of transposition efficiency was reduced to approximately 35% of the wild-type transposase (Figure 2B,D). These results indicated that the disordered region plays an essential role in transposition: its deletion significantly reduces the transposition rate to varying extents in different human cell types.

### 2.2. The Sequence of NTDR Is Important for Its Proper Function

To determine whether the amino acid sequence is also important for the functional influence (in addition to the disordered feature of the region), we replaced the original NTDR with segments bearing similar disordered characteristics but having a randomized amino acid sequence. This approach allowed the preservation of the amino acid composition but disrupted any function that may be linked to specific sequence motifs. We tested two scrambled NTDR PB mutants named mPBvar14 and mPBvar73. We performed the transposition assays with these variants in all three cell lines described above to evaluate if these sequences can complement the NTDR deletion phenotype. These experiments showed the inability of these disordered protein segments to rescue wild-type transposition efficiency; in most cases, it was even further reduced as compared to the mPB-del activity (Figure 1 and Figure 2) except for mPBvar14 in MCF-7 cells, where the transposition activity was slightly higher than for the mPBdel mutant (Figure 2D). The results support the hypothesis that sequence conservation of the PB transposase NTDR is important for its functionality. In addition, since the deletion or the replacement of this disordered segment did not abolish transposition, it indicates that this region contributes to the “fine-tuning” of the reaction, most likely via interacting with cellular partners.

### 2.3. The Effect of NTDR Deletion on the Hyperactive PB Variant

To further investigate the functional role of the NTDR, we deleted this region also from a recently developed hyperactive PB variant (hyPB, [18]) (Figure 3A). In line with the previous findings for mPB, transposon excision was always detected (Appendix A). However, in contrast to the results for the mPB transposase, no significant changes were detected between transposition efficiencies of the hyPB and the hyPBdel variants in HEK-293 or HeLa cells (Figure 3B,C). On the other hand, in MCF-7 cells, we detected an approximately 40% reduction in the overall transposition rate for the hyPBdel variant, again indicating cell type-specific regulation of transposition (Figure 3D). When the NTDR was replaced with either of the previously generated, randomized sequence disordered segments (denoted as hyPBvar14 and hyPBvar73), we detected a substantial reduction of transposition in all three cell lines examined, with the most prominent effect seen in HeLa cells with the var73 sequence (Figure 3B–D).

There is only one amino acid difference in the NTDR of hyPB compared to mPB. To determine how much this I30V mutation contributes to the transposition efficiency, we mutated this amino acid position back to the original isoleucine found in the mPB sequence (the mutant is denoted as hyPB^30I^, see Figure 3A). This “back mutation” showed a similar cell type-specific effect on transposition as the hyPBdel: no significant change or a slight tendency of increase was detected in the HEK-293, and HeLa cell lines, respectively, whereas a more minor but significant decrease was seen in the MCF-7 cell line (Figure 3B–D). All these results further supported a cell type-specific role of this N-terminal region in the transposition function. They indicated the importance of sequence constraint in the NTDR of PB transposase.

### 2.4. NTDR Deletion Has a Higher Impact on the Excision Step of Transposition

In an attempt to distinguish the influence of PB NTDR on the distinct steps of transposition, we used a specific reporter cell line to assess the deletion of this region during the excision and on the integration step. This HeLa-derived cell line contains one copy of a construct where a puromycin resistance gene is interrupted by a PB transposon carrying a neomycin resistance cassette (Figure 4A). Suppose a functional PB transposase is expressed in this cell line. In that case, it could tracelessly remove the transposon resulting in puromycin resistance, thereby quantitatively scoring the efficiency of the excision step of the reaction. In addition, if a puromycin-neomycin double selection is applied, one can screen for the complete transposition reactions ending with functional integration events. We compared the activity of the mPB transposase and the mPBdel variant in this reporter system and detected a significant reduction, by approximately 70%, in the transposon excision by mPBdel (Figure 4B, upper panels, compare the ‘exc’ values for mPB and mPBdel). The overall transposition efficiency (number of the combined excision and integration events) also drops significantly due to the NTDR deletion. However, the ratio of integrations among the excised transposon population does not decrease: its value is higher for the mPBdel variant (Figure 4B, upper right graph). These results indicate that the excision step during PB transposition is influenced more significantly by the removal of NTDR than the subsequent step of genomic integration.

We also performed similar experiments on this HeLa reporter cell line with the hyPB variants. It was revealed that in such limited substrate conditions (acting on a single transposon copy in the genome), the hyPB transposase was more effective than the mPB version, both in the absolute numbers of excision events, as well as in the integration rates, as expected. On the other hand, as compared to the mPBdel mutant, the NTDR deletion of hyPB resulted in a more severe (reaching 90%) reduction in excision efficiency. However, the integration rate did not decrease for the hyPBdel mutant: in fact, it also showed a tendency to increase, similarly to the mPBdel mutant (Figure 4B, graphs on the right). Again, these results further support the previous observation that the excision step of transposition is more significantly affected by a deletion of the NTDR in the PB transposase.

### 2.5. Conservation of the NTDR in Domesticated PB-Derived Proteins

Seeing that the NTDR is necessary for the proper function of mPB, we asked whether the presence of such a longer disordered region at the N-terminus is a characteristic feature of other *piggyBac*-related proteins. Using the IUPred2 program [45,46] to predict the disorder tendencies of the five domesticated human *piggyBac*-derived sequences (PGBD1-5) [22,31], we found that all of these proteins contain a long stretch of structurally disordered regions upstream of the transposase domains, even PGBD1, which has gained additional N-terminal protein domains during evolution (Figure 5A). Inspired by this, we conducted a further systematic analysis of all the currently available sequences of this superfamily of DNA transposons. This included active transposases and domesticated PB proteins that are proven or predicted to have become domesticated. This analysis revealed that more than 95% of PB proteins contain a disordered stretch of >10 amino acids upstream of the transposase catalytic domain and can be considered NTDR (Appendix A). We conclude that such a protein region is a widespread feature of these transposase-originated proteins, even those that have acquired additional protein domains during evolution. Based on their sequences and their domain structures, the analyzed proteins could be classified into distinct groups: most of these sets could be defined and named after the five representative domesticated human PB proteins, PGBD1-5; in addition, proteins that do not fit into those categories form an extra group (these are almost exclusively active transposases, including the *T. ni* insect transposase). Most of the members belonging to the *piggyBac1* and the *piggyBac4* groups contain extra domains: the majority of the former group having acquired a SCAN and a KRAB domain during evolution which is located upstream of the transposase domain [47]. As opposed to this, the *piggyBac4* proteins contain a Zn-ribbon domain downstream of the catalytic domain of the transposase. Despite these extra domains, all these PB proteins still have a conserved NTDR upstream of the catalytic domain of the transposase, indicating its functional relevance even for the domesticated functions. However, it is important that the evolutionary most ancient *piggyBac5* group contains the most members where such NTDR cannot be recognized (Appendix A).

It is also important to note that the Anchor algorithm, designed to identify potential interaction sites within disordered sequences [48], shows a high probability of the NTDR regions containing such regions (Figure 5A, blue lines). This underlines the relevance of the experimental results, where the scrambled NTDR sequences could not restore the transposase activity of mPB. In agreement with this observation, the scrambled variants do not have a pronounced disordered binding site, according to the ANCHOR prediction (Appendix A).

Finally, we addressed the question of whether an NTDR is also present in other DNA transposases from different superfamilies. We performed a disorder prediction for the *Tol2* transposase from the hAT superfamily and the *Setmar* protein from the Tc1/Mariner superfamily. In contrast to the PB proteins, they do not show an N-terminal concentration of disordered region; the predictions indicated the absence of such long disordered regions throughout the entire proteins (Figure 5B). These results supported the previous observation that the presence of a long NTDR upstream of the catalytic domain of the transposase is a characteristic feature of the PB superfamily.

## 3. Discussion

Although various domains of the PB transposase have been mapped and functionally well characterized, the N-terminal disordered region has not been examined, and its function remains elusive [39]. In this study, we investigated the role of this NTDR by performing transposition reactions with various deletion mutants in different cell lines. Testing the mPB transposase lacking the NTDR, we could detect a drastic and significant drop in transposition efficiency in a strong cell type-dependent manner. However, transposition was not completely abolished, indicating that the presence of the NTDR is not essential for transposition. Interestingly, suppose the NTDR was replaced with disordered protein segments with the same amino acids in a randomized order. In that case, this could not complement the deletion phenotype, but it further lowered the transposition rate in most cases. These results clearly showed that the sequence of the NTDR also carries functional relevance, possibly through interaction with other molecules, as suggested by the Anchor predictions.

As opposed to the results with mPB, deletion of the NTDR in the hyperactive hyPB transposase [18] did not recapitulate the negative phenotype in most cell lines, whereas the scrambled mutants always severely lowered transposition efficiency. The results indicate that the 7 “hyperactive” mutations could collectively compensate for the lack of NTDR in a cell type-specific manner while simultaneously sensitizing the protein to the presence of a dysfunctional disordered segment. The inhibitory effects of cell-type specific factors cannot be ruled out at this point, but so far, no reliable information on those is available. The hyPB was initially developed in a yeast system, with subsequent testing in mouse embryonic stem cells. Although significantly outperforming the wild-type mPB variant, substantial differences in the activity were detected even between those cell types [18]. In a later systematic study, the robustness and the clear advantage of the hyPB were demonstrated in human embryonic stem cells. In contrast, in other cell types, it showed significant variability, not always providing the expected “hyperactivity” [49]. Such cell type-dependent variability was demonstrated here with the hyPB NTDR deletion mutant. We could show that the only “hyperactive” mutation (I30V) positioned in the NTDR provides only a minor contribution to the hyperactive phenotype and, therefore, to the lack of transposition impairment of the NTDR-deleted hyPB. Nevertheless, further systematic studies are needed to investigate the exact structural role of these mutations and to decipher how and to what extent they can override the molecular interaction(s) of the NTDR sequence.

Our results with a single copy reporter cell line revealed that the excision step of transposition is more severely affected by the NTDR deletion, both for the mPB and the hyPB transposase variants. Previous studies already indicated that the two stages of transposition are kinetically distinct: the excision efficiency of the hyPB showed a 17-fold increase compared to the wild-type mPB variant, whereas the integration efficiency was elevated “only” by 9-fold [18]. A definite proof was provided by developing an integration deficient PB transposase, showing the structural background for the selective inhibition of the integration step [50]. Future studies would be needed to see how the NTDR deletion affects this variant. Such experiments would help focus on the selective function of the disordered region on the PB excision reaction.

Our bioinformatic analyses provided evidence that such a disordered protein region is present at the N-terminal part of the transposase domain in other *piggyBac*-related proteins. It was revealed that the position of this region is well-conserved among the members of the *piggyBac* superfamily, even among the domesticated proteins that were shown to have lost their ability for transposition, including the five domesticated human PGBD proteins [22]. It is intriguing that regardless of the acquisition of other non-transposase domains during evolution (as in the case of PGBD1 or PGBD4), the position of the disordered region is strictly kept upstream of the DDE_Tnp_1_7 transposase domain among *piggyBac* proteins; however, selected members from other DNA transposon superfamilies do not show this structural feature. This special structural constellation of the *piggyBac*-related sequences would further point to functional conservation, but the question remains: what would be the role of NTDR in PB proteins that do not show excision activities anymore [22,47]? A possible explanation could be that certain protein–protein interactions may well serve the new domesticated functions, and understanding the role of the NTDR in PB transposition would therefore be helpful to reveal the endogenous function of the co-opted proteins.

So, what could be revealed from these experiments concerning the potential function of the N-terminal disordered region? The most likely conclusion is that it is involved in various protein–protein interactions that modulate the transposition reaction. The PB transposase does not seem to require many protein partners, indicated by the lack of strict host specificity and the wide range of organisms where this transposon system can be used. However, there are certain protein groups that the transposase possibly interacts with: these include the DNA repair factors to suppress genotoxic effects caused by double-stranded breaks generated during transposition or certain chromatin-associated factors that determine the distinct integration profile of the transposon [23,51]. Cell cycle regulating proteins may also be good candidates as such interactions were shown to be important for controlling other DNA transposases, connecting their activities to cell proliferation [52]. Our preliminary attempt to validate selected DNA repair or cell cycle factors as potential *piggyBac* regulators failed. The results indicated that a future systematic screen is required to identify *bona fide* NTDR interacting partners. The flexibility of a disordered region could provide a platform to interact with several protein partners or even different parts of the same protein, promoting different functions on the same partner, referred to as the “moonlighting function” of disordered proteins [44]. Previous studies showed that the lack of disorder-to-order transition, alternatively called the “fuzziness” of intrinsically disordered proteins [53,54], could provide the structural flexibility to perform unrelated functions on different substrates, as proven by the versatile actions of the thymosin-β4 protein [55]. Another example showed that a disordered segment could have opposing functions on the same target protein. As a result of two independent stochastic actions, a random-coil fragment of the dihydropyridine receptor can activate or inhibit its ryanodine receptor target during muscle contraction [56]. Based on such examples, one can hypothesize that the NTDR could enhance the excision step with specific protein partner(s). Subsequently, it may also promote substrate integration during PB transposition.

Another possibility is that the NTDR can modulate interactions between individual PB transposase proteins or even mediate intramolecular connections to initiate “higher order PB assembly synapses”, as predicted by Chen et al. [39], and the CRD region of the *piggyBac* protein (Figure 1B) may be a good candidate for that [41]. However, our results on the cell-type specific behavior of the NTDR mutants point to the role of interaction with other protein partners, whose cell-type specific expression profiles may lie behind the observed differences. Nevertheless, it would be interesting to see if any revealed functions are still conserved in the domesticated PGBD proteins or if new functions arose for the NTDR in those cases. Another proposed function of disordered protein segments is that they may facilitate protein evolution via exon shuffling by structurally separating the newly combined domains [57]. One can argue that this could be the case for mammalian PGBD1 proteins [47]. However, it does not exclude other functional role(s) of the conserved NTDR. Be that as it may, further experimental studies are required to decipher the potential function(s) of the N-terminal disordered region in *piggyBac* transposition, which results would help elucidate the domesticated functions of various co-opted *piggyBac*-derived protein sequences.

## 4. Materials and Methods

### 4.1. Plasmid Constructs

Structures of donor and helper plasmids used for the transposon assays are shown in Figure 1A. Donor plasmids contain *piggyBac* transposon units carrying either a PGK promoter-driven puromycin resistance gene or a CAG promoter-driven GFP gene [49]; the helper plasmids contain a CMV promoter-driven appropriate *piggyBac* transposase variant expression cassette (see the figure legend for more details). The codon-optimized mPB [13] and the hyperactive hyPB [18] transposases were used as positive controls, and the D268G catalytic mPB mutant generated earlier [22] was used as a negative control. All other expression plasmids used in these studies were generated by the Gibson assembly method and verified by Sanger sequencing; primers used for the assemblies are listed in Appendix A. For the NTDR deletion mutants mPBdel and hyPBdel, amino acids 3–100 from the N-terminal part of the proteins are removed. For variants mPBvar14, hyPBvar14, mPBvar73, and hyPBvar73, the disordered sequences ‘14’ (VNHPSHDLSSSETIVDADETDLQLDWKSDLELHDTCNVDEETISSQVNLKGEIDGVPLTQDSIGRQFPSASILSDVEGIQSEISLREESEDQSREASH) and ‘73’ (SVATATLNIKDSDHGPVEDIQERARTDSCSSVELDEDDEQIELDHGVIDSSHLQLQLDWEGNLELPQTFSISKHSSEVSDTGQLRSSSNVEIIDEPES) were used to replace the amino acids 3–100 of the original transposases, respectively. These sequences were generated from the original sequence by randomizing the amino acids, resulting in the same amino acid composition and similar disorder characteristics. Coiled-coil containing aggregation-prone sequences were filtered out. Since the I30V mutation of hyPB is localized in the NTDR, for comparative studies, this position was mutated “back” to the original isoleucine present in mPB, thus generating the hyPB30I variant. For the HeLa reporter cell line (see Figure 4), a single genomic copy of the PB transposon disrupting a puromycin resistance gene cassette was stably inserted using the *Frog Prince* transposon system [58].

### 4.2. Cell Culturing and Transfection Methods

HEK-293 (human embryonic kidney), HeLa (human cervical cancer), and MCF-7 (human breast cancer) cell lines were cultured in Dulbecco’s modified Eagle’s medium (DMEM) supplemented with 10% of fetal calf serum, 1% of L-glutamine, and 1% of penicillin–streptomycin (Thermo Fisher Scientific, Waltham, MA, USA).

Transfections were carried out in duplicates. For the lipid-based transfections, 5 × 10^5^ cells were seeded onto 6-well plates. The next day, 500 ng transposon donor and 500 ng transposase expressing helper plasmid were co-transfected to the cells using the FuGENE^®^ 6 reagent, according to the manufacturer’s instructions (Roche Applied Science, Mannheim, Germany). At 48 h post-transfection, cells were harvested for further experiments or analyses.

### 4.3. Transposition Assays

Transposition assays were carried out essentially as described previously [22]. Briefly, for excision assays, plasmids were isolated from the transfected cells using a modified protocol of the QIAGEN plasmid Miniprep Kit, then 10 ng of the isolated plasmids were used as templates in a two-round nested PCR assay to detect plasmid copies that underwent transposon excision and DNA repair; the ampicillin sequence presenting in all donor construct was used as an assay control. PCR products were separated in a 2% agarose gel and visualized by ethidium bromide staining using a Universal Hood Gel Imager Model # 75S (BioRad, Hercules, CA, USA). PCR primers for excision analysis are listed in Appendix A.

For colony assays, starting at 48 h post-transfection, 1% of the transfected cells were seeded onto cell culture Petri dishes and selected for two weeks using puromycin (Sigma-Aldrich) in the final concentration of 1 µg/mL. Surviving cells were fixed in ice-cold methanol and stained with 0,04% Crystal Violet (Sigma-Aldrich, St. Louis, MO, USA) in 25% methanol. To test the effects on distinct steps of transpositions, we used a specific HeLa-derived reporter cell line (see above and Figure 4). These cells were treated with 100 mg/mL neomycin and 1 µg/mL puromycin simultaneously for double antibiotic selection. After the selection procedure, surviving colonies were quantified with the Universal Hood Gel Imager Model # 75S, using the Quantity One 4.4.0 software (BioRad). For statistical analyses, the mean and standard deviation values of at least three independent biological replicates were compared, performing F-tests and subsequent two-sided Student’s *t*-tests; statistically significant differences were accepted at *p* < 0.05.

### 4.4. Analyzing piggyBac Sequences and Disorder Predictions

We created a nonredundant *piggyBac* protein dataset by an advanced search in the UniProtKB database (Release 10 February 2021) with the following keywords: protein name: ‘piggybac’ OR protein name: ‘dde_tnp_1_7’ OR gene name: ‘pgbd’. The resulted dataset of 4277 proteins was narrowed down: first, proteins that couldn’t be classified, based on protein or gene name, as type 1, 2, 3, 4, or 5 were removed. Next, the dataset was narrowed down to have a maximum of 1-1 protein per species and type, the longest form was kept from each *piggybac* type. Minor corrections were done based on the Pfam database (http://pfam.xfam.org/) (accessed on 13 May 2021 and 18 January 2022), and proteins without transposase domain were removed. The protein classification was also corrected based on the extra domain content if needed. *Piggybac* type 1 proteins often contain a SCAN domain, while many *piggybac* type 4 proteins contain a Zn-ribbon domain. Some known and important *piggybac* proteins without classification were added to the list: ‘*piggyBat*’ (*Myotis lucifugus*), not in UniProt, translated from DNA; ’*piggyMac*’ (*Paramecium tetraurelia*), A0DFJ7; piggybac-like protein Tpb2p (*Tetrahymena thermophila*), D2Z1K6; Transposase mPB (*Trichoplusia ni*), Q283G1. Based on its Zn-ribbon domain, *piggyBat* was classified as a type 4 *piggybac* protein.

For control datasets, we used 2 transposase families: Tc1 mariner transposase family and *Tol2* transposases. We carried out an advanced search in UniProtKB with the following keyword: gene name: ’setmar’ (SETMAR: SET domain and mariner transposase fusion protein homolog) to find Tc1 mariner transposons and narrowed down the dataset as we did in the case of *piggybac* proteins. Such a comprehensive investigation in the case of *Tol2* transposases could not be completed because not enough candidates were found in Uniprot. A known *Tol2* transposase is Q9PTV1.

Structural disorder of transposase proteins was predicted by the IUPred algorithm (https://iupred2a.elte.hu/) (accessed on 20 May 2021 and 25 January 2022) [46], which is based on estimating the total pairwise inter-residue interaction energy gained upon folding of a polypeptide chain. The predictor returns a position-specific disorder score in the range of 0.0–1.0, and an amino acid with a score at least 0.5 is considered locally disordered. The mean disorder was computed as the average of residue scores. To characterize the disorder tendency of the whole protein or different parts of the protein (domains, 100 amino acid long regions before the transposase domains, regions before and after the transposase domains), we calculated the ratio of disordered residues within the given region. Domain boundaries are derived from Pfam database (http://pfam.xfam.org/) (accessed on 13 May 2021 and 18 January 2022).

## Figures and Tables

**Figure 1 ijms-23-10317-f001:**
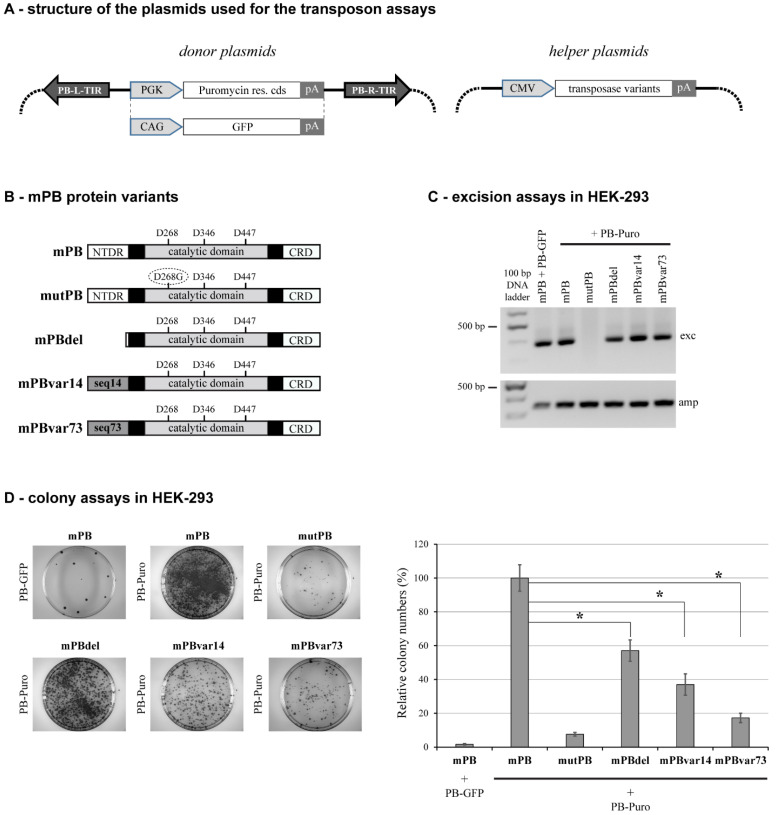
Testing the activities of *piggyBac* transposase mutants in HEK-293 cells. (**A**) Schematic overview of the donor and helper plasmids used for the excision and the colony assays. PB-L-TIR/PB-R-TIR: *piggyBac* left/right terminal inverted repeats; PGK: phosphoglycerate kinase promoter; CAG: cytomegalovirus-actin-globin artificial hybrid promoter; CMV: cytomegalovirus promoter; Puromycin res. cds: Puromycin resistance coding sequence; GFP: green fluorescent protein; pA: polyadenylation signal. (**B**) A schematic representation of the *piggyBac* transposase protein variants. In the catalytic domain, the positions of the three important aspartic acids (D) are shown; for the catalytic mutant (mutPB), the 268 position is mutated to glycine (G) as indicated. The two black boxes represent the two distinct regions of the ‘dimerization and DNA-binding domain’. NTDR: N-terminal disordered region; CRD: cysteine-rich domain; seq14/seq73: two scrambled protein sequences designed to have similar disordered characteristics as the original N-terminal region of mPB while preserving the amino acid composition. (**C**) A representative result of the diagnostic PCRs detecting the occurrence of transposon excision. The upper product (‘exc’, 381 bp) is derived from the excised and repaired donor plasmids. The lower product (‘amp’, 340 bp) is a control amplicon from the ampicillin resistance gene on the backbone of the transfected plasmids. (**D**) On the left, representative images of plates are shown containing resistant colonies after puromycin selection (digital photos of Petri dishes with diameter of 10 cm). The graph on the right presents the quantification of colony numbers for each assay condition. All values are expressed as percentages of the ‘mPB+PB-Puro’ control reaction. Mean values of at least four independent biological experiments are shown, and error bars represent standard deviations, *: *p* < 0.05. For excision and colony assays, the combinations of donor and helper plasmids are indicated for each reaction.

**Figure 2 ijms-23-10317-f002:**
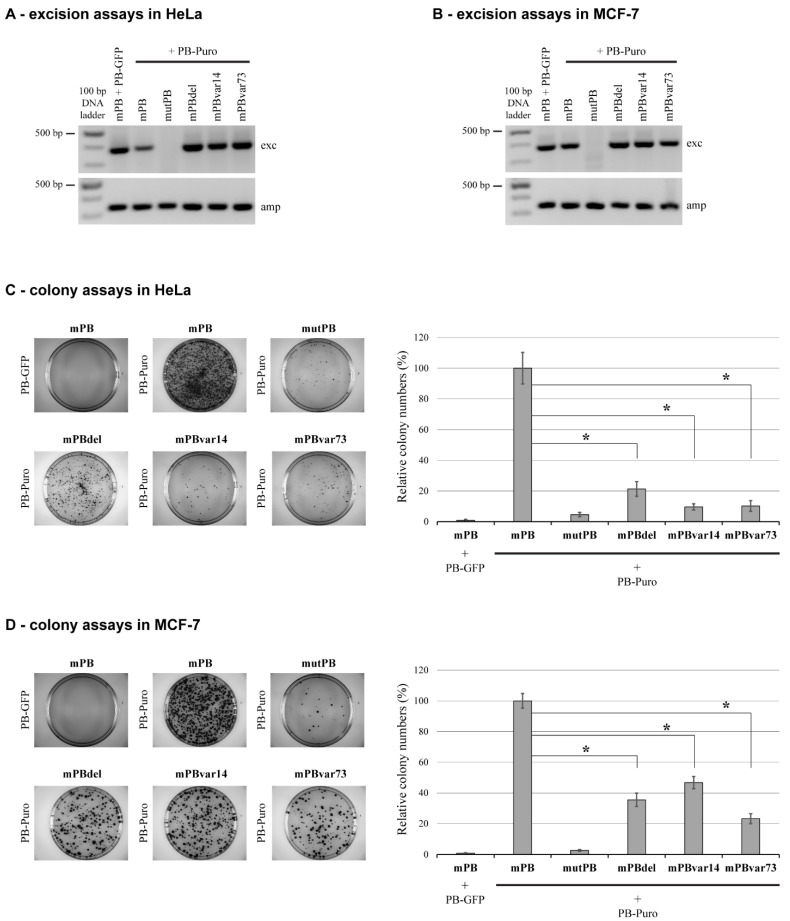
Testing the activities of *piggyBac* transposase mutants in HeLa and MCF-7 cells. (**A**,**B**) show representative images of excision PCRs in HeLa and MCF-7 cells, respectively. The upper band (‘exc’, 381 bp) is detected when transposon excision and plasmid repair have occurred, whereas the lower control band (‘amp’, 340 bp) is detected in case of successful plasmid transfection. (**C**,**D**) show representative images and quantification of colony assays in HeLa and MCF-7 cells, respectively (digital photos of Petri dishes with diameter of 10 cm). Shown values are means of at least six (HeLa) or three (MCF-7) independent biological replicates. Error bars represents standard deviations, *: *p* < 0.05.

**Figure 3 ijms-23-10317-f003:**
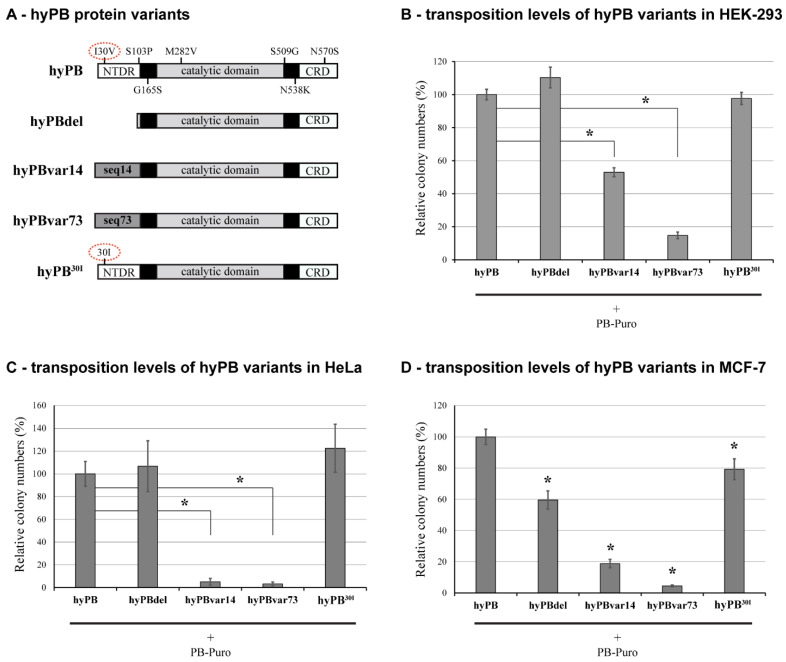
Testing the activities of the hyperactive *piggyBac* transposase mutants in three cell lines. (**A**) A schematic representation of the hyPB transposase protein variants. The hyPB differs from the mPB protein in 7 amino acid positions indicated on the upper drawing. Only one of these positions (indicated in red dashed lined ovals) is affected by the NTDR deletion (I30V) and tested for its influence on hyPB activity (30I). The domain structure of the transposase is indicated in Figure 1B. (**B**–**D**) panels show quantifications of colony assays in HEK-293, HeLa, and MCF-7 cells. Shown values are means of at least three independent biological replicates. Error bars represents standard deviations, *: *p* < 0.05.

**Figure 4 ijms-23-10317-f004:**
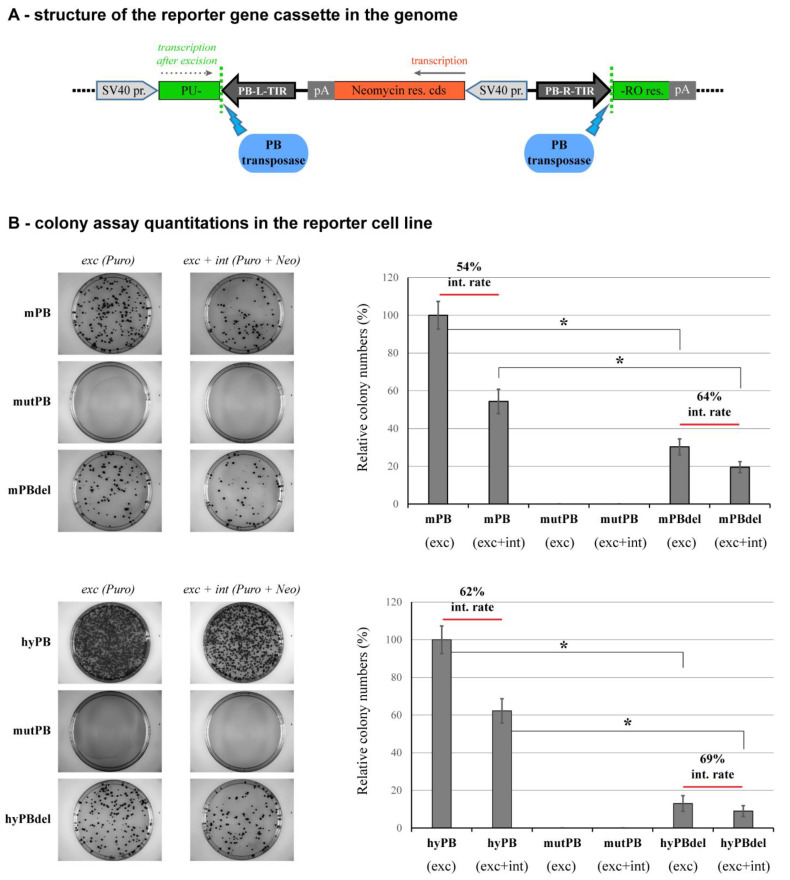
Testing the activities of the mPB and the hyPB transposase variants in a HeLa reporter cell line containing a single copy of a transposon substrate. (**A**) Structure of the reporter gene cassette in the genome. A puromycin resistance gene (marked in green) is interrupted with a *piggyBac* transposon carrying a neomycin resistance gene cassette (marked in red). If an active transposase is expressed in the cells (depicted by a blue oval), it may seamlessly excise the transposon unit, thereby restoring puromycin resistance. SV40 pr.: simian virus 40 early promoter; PB-L-TIR/PB-R-TIR: *piggyBac* left/right terminal inverted repeats; pA: polyadenylation signal. (**B**) On the left, representative images of plates are shown after selection with antibiotics (digital photos of Petri dishes with diameter of 10 cm). Puromycin treatment selects only for transposon excisions (‘exc’), whereas puromycin and neomycin double treatment selects for transposon excision and integration events (’exc+int’). On the right, the graphs show the quantification of colony numbers for various assay conditions. All values are expressed as percentages of the mPB (upper graph) or hyPB (lower graph) control reactions; the experiment is carried out with the catalytically inactive transposase mutant for negative control. Integration rates (‘int. rate’) for double selections are calculated as the percentage of mere excision events (puromycin selection) for each transposase variant. Mean values of at least three independent biological experiments are shown, and error bars represent standard deviations; *: *p* < 0.05.

**Figure 5 ijms-23-10317-f005:**
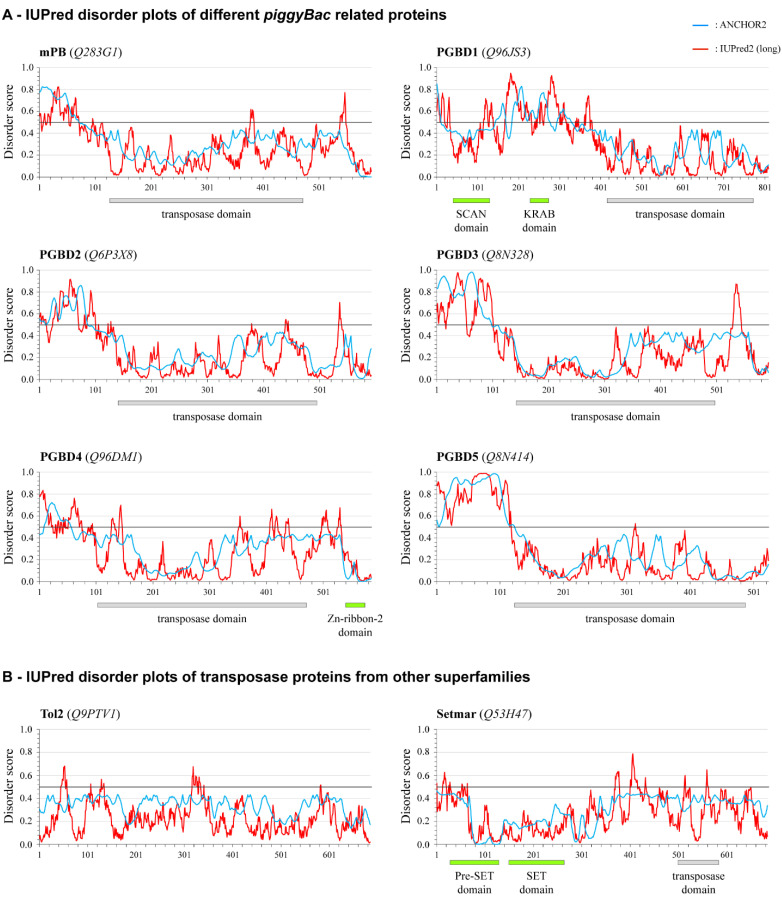
Prediction of disordered properties of various transposase proteins using the IUPred prediction software. (**A**) Disordered plots of the mPB transposase and the five human *piggyBac*-derived domesticated protein sequences (PGBD1-5). Amino acids numbered from the N-terminal of the proteins are shown on the *x*-axis; disorder scores plotted on the *y*-axis, calculated using the ANCHOR2 (blue line) or the IUPred2 (red line) algorithms. A peptide region is considered structurally disordered if the score values are above 0.5, and the horizontal solid black line indicates this threshold. (**B**) Disordered plots of non-*piggyBac* transposase proteins: *Tol2* belongs to the hAT superfamily, whereas *Setmar* belongs to the Tc1/Mariner superfamily. Known domains (transposase or others) are indicated under the graph of a given protein; UniProt identifiers of proteins are shown in parenthesis.

## Data Availability

Not applicable.

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
