# Peer review of "Functional Characterization of the N-Terminal Disordered Region of the *piggyBac* Transposase"

_ijms, 2022, doi:10.3390/ijms231810317_

Round 1
Reviewer 1 Report
The present study functionally characterizes the N-terminal domain of PiggyBac transposases using molecular techniques and bioinformatic analysis. The article is well-written, and the experimental design is scientifically sound. The focus of the study is relevant to our understanding of transposition events in humans. Below are some recommendations to help improve the current version of the manuscript:
As general comments:
1- A major concern in the current version is the high variability between models in response to NTDR mutation. The current conclusions need to be strengthened by either identifying the cause of cell-specific responses or including more models to determine the most common trends.
2- The gels need to be quantified and normalized to the control input (Amp).
3- The manuscript focuses on the N-term which appears to have some modulatory functions. However, besides the catalytic domain which was mutated as a negative control, the CRD domain has not been taken into consideration in the experiments. Testing the interference between the NTDR and the CRD is highly relevant and should be tested.
Specific comments:
1- The high variability in response to NDTR mutation remains a concern and needs to be addressed. One option would be to evaluate whether cell proliferation differences between various models influence the transposition activity. Cell cycle regulators could be used to module cell proliferation. Alternatively, more cell lines with different kinetics may be included. The fact that the hyPB reveals less change in transposition despite NTDR deletion further supports the need to assess cell proliferation in the study.
2- The authors suggest that cell-specific factors could be recruited to promote transposition. Experiments could be performed to determine whether different factors are indeed recruited in control mPB and or mPBdel.
3- The experiment performed in Figure 4 to evaluate the excision versus the transposition activity of NTDR del was very clever. In figure 4B, hyPB was presented without a mutant control for the catalytic domain. Consider updating for background purposes.
Reviewer 2 Report
In this manuscript (ijms-1854001) ” Functional characterization of the N-terminal disordered region 2 of the piggyBac transposase “ Wachtl, et al., tried to demonstrate the important of the N-terminal disordered region (NTDR) of transposase, piggyBac that derived from cabbage looper (Trichoplusia ni). They used the deletion and sequence scrambled as well as bioinformatics approach to identify the functional importance of NTDR of piggyback. These results are solid and will imply the future studies of the functional partners of the transposase. Some minor comments are listed below:
1. Line 20: the protein also contains a ~100 bp N-terminal disordered region (NTDR), ~100 bp must be changed to amino acids!
2. A figure that illustrated the sequence comparison about the NTDR among the
piggyBac , hyperactive PB variant and other transpoase may be required.
3. In Figure 1., the promoters, like CMV or CGA, an “arrow” may be better than the “block”? and puromycin res. cds should address it.
4. Why the hyperactive PB variant did not require the NTDR in HEK-293 cells and HeLa cell lines? I am interesting to know whether this phenomena also appear in the insect cells, the origin of the piggyBac derived
Round 2
Reviewer 1 Report
Most experiments suggested by the reviewer are either ongoing or non-conclusive. These limitations have been discussed in the revised version without any supporting evidence. Despite my enthusiasm for this manuscript some of the concerns need to be addressed to grant publication. The control experiment for hyPB in Figure 4B is not relevant to that panel and is just the duplicate of the top. The minimum would be to include a mutant hyPB not mPB.
Author Response
>Most experiments suggested by the reviewer are either ongoing or non-conclusive. These limitations >have been discussed in the revised version without any supporting evidence. Despite my enthusiasm >for this manuscript some of the concerns need to be addressed to grant publication. The control >experiment for hyPB in Figure 4B is not relevant to that panel and is just the duplicate of the top. The >minimum would be to include a mutant hyPB not mPB.
We respectfully disagree with the raised comment. A catalytically inactive transposase mutant is a relevant control in this experiment as it shows the complete lack of excision and integration of the transposon in the genome, and the clear background after selection (as shown on the plates). We clarified this issue by explaining the control experiment in the legend to Figure 4 (lines 260-261 in the revised manuscript). The hyPB transposase is a mutated version of the original mPB, similarly to the mPBdel and the hyPBdel mutants, and if the catalytic sites were mutated in all versions, they would all be inactive and show no transposase activity at all.
We would like to thank the Reviewer for her/his comment and we hope that s/he finds the revised manuscript acceptable for publication.
Round 3
Reviewer 1 Report
The revised version remains unsatisfactory, but this may serve as a basis for more studies to come.